# Effects of Different Drying Methods on the Quality of *Bletilla striata* Scented Tea

**DOI:** 10.3390/molecules28062438

**Published:** 2023-03-07

**Authors:** Xue Han, Zhiqin Song, Jiawei Liu, Yeshan Zhang, Mingkai Wu, Hai Liu

**Affiliations:** 1Institute of Modern Chinese Herbal Medicine, Guizhou Academy of Agricultural Sciences, Guiyang 550002, China; 2Guizhou Institute of Crop Variety Resources, Guiyang 550006, China; 3Guizhou Key Laboratory of Agricultural Biotechnology, China Guizhou University, Guiyang 550006, China

**Keywords:** *Bletilla striata* scented tea, gas chromatography–mass spectrometry (GC-MS), relative odor activity value (ROAV), drying method, quality

## Abstract

Flower tea is widely loved as a drink, especially for the beautiful and rich flowers of the orchid family, and the drying method for different flowers is also unique. GC-MS is widely used to study volatile substances to determine the quality of flower teas. The findings show that the freeze-drying method can retain the original aroma and flavor of *Bletilla striata* has the highest sensory evaluation score, with the key flavor substances ethyl caproate and *N*-heptanal containing 1.14% and 6.28%, respectively, and their ROAV values reaching 54.46 and 100.00. Additionally, the freeze-drying method can well retain flavonoids, polysaccharides, and phenolic components, while providing better antioxidant and antibacterial properties. The stove-drying method would make *Bletilla striata* slightly burnt and less flavorful and efficacious than freeze-drying; the air-drying method is difficult to retain the special odor and fragrance of *Bletilla striata* flowers and has the lowest sensory evaluation score, with the presence of volatile components with irritating and unpleasant odors such as pyrazine and 2-pentylfuran, while not showing better efficacy. In addition, steam fixation would destroy the morphology and flavor of *Bletilla striata*, lose polysaccharide and phenolic components, and reduce the efficacy of *Bletilla striata* scented tea, but could retain the flavonoid components well. In summary, direct freeze-drying without steam fixation is the best process for drying *Bletilla striata* scented tea, stove-drying without steam fixation is more economical and convenient in actual production and application, steam fixation and air-drying are not suitable as drying processes for *Bletilla striata* scented tea. This study analyzed the quality of *Bletilla striata* scented tea under different drying methods, promoted the further processing of *Bletilla striata* scented tea, and provided a reference for the comprehensive utilization of *Bletilla striata* scented tea.

## 1. Introduction

*Bletilla striata* (Thunb.) Reichb. f. is a rare Chinese medicinal herb of the orchid family in southwestern China, with the unique fragrance and flowering period of April to June. The flowers are light purple, which not only have high ornamental value, but also have certain pharmacological activities [1,2]. At present, the planting area of *Bletilla striata* in Guizhou Province is more than 78 km2, and the yield of *Bletilla striata* flowers can exceed 60,000 kg per square kilometer [3]. *Bletilla striata* scented tea has antibacterial, antioxidant, and other characteristics, and has potential anti-aging and antihypertensive activities [4].

Due to the unique fragrance and flavor, *Bletilla striata* scented tea is widely accepted by people. Studies have found that the *Bletilla striata* flowers are rich in phenols, flavonoids, polysaccharides, and volatile components [5,6], which are considered to have antioxidant, antibacterial, anti-tumor, and immunity-enhancing activities [7,8,9]. *Bletilla striata* flowers are usually subjected to drying before transportation and storage. However, the capacity and technology of drying *Bletilla striata* flowers are insufficiently developed, which depends on the experience of self-harvesting and processing by local farmers. Without scientific guidance, such a raw drying process usually causes a waste of flowers, low quality, bad taste, and unstable batch-to-batch consistencies of scented tea. Some other drying methods like freeze-drying and stove-drying were also used in processing scented tea that better retain the natural fragrance, good taste, and potential therapeutic efficacy [10]. However, the internal relationship between the drying method and the quality of scented tea is rarely studied. Therefore, it is urgent to study the impact of different drying methods on the quality of *Bletilla striata* scented tea.

This study explored the effects of different drying methods on *Bletilla striata* scented tea from three aspects: sensory, component, and effect. *Bletilla striata* scented tea after different drying methods was analyzed through sensory evaluation and gas chromatography–mass spectrometry (GC-MS). A variety of chemical substances, such as total flavonoids, polysaccharides, phenols, antioxidants, and antibacterial agents, was determined to reveal the relationship between *Bletilla striata* scented tea quality and different drying methods. This study advanced in-depth research and comprehensive utilization of *Bletilla striata* scented tea, which provided important practical guidance for its industrial production.

## 2. Results and Discussion

### 2.1. Effect of Different Drying Methods on Sensory Evaluation of Bletilla striata Scented Tea

According to the sensory evaluation method of tea, the *Bletilla striata* scented tea obtained under different drying methods was brewed, and the results are shown in Figure 1. From the color, shape and aroma of scented tea, No. 1 and No. 2 both retained the original fragrance of *Bletilla striata* flowers in varying degrees, but No. 2 had some discoloration. Although No. 3 retained the original form of scented tea to some extent, its color faded to a brownish red. The color of No. 4 was slightly faded to light reddish brown, its flower shape was not good, and its fragrance was slightly light. No. 5 and No. 6 were close to each other, showing a black tea block with no fixed shape, and the special aroma of *Bletilla striata* flowers was not enough, but No. 5 showed a green tea aroma slightly, while No. 6 had no evident tea aroma. From the perspective of color, aroma, clarity, and taste of the tea soup, the No. 1 tea soup had a stronger aroma than the No. 2 tea soup, which better retained the original special aroma of the *Bletilla striata* flowers, and the aroma did not dissipate after brewing for 30 min. The No. 3 tea soup was yellow–brown, relatively clear, but the taste was not good, and the leaf bottom was relatively stretched, showing light brown. The No. 4 tea soup was pink, slightly light in aroma, slightly light in taste, and the leaf base was not stretched enough, showing a light pink color. The No. 5 tea soup was brownish red with a slight caramel flavor. After brewing for 30 min, the aroma was not evident, and the taste of the tea soup was not good. No. 6 was brownish yellow, slightly red, slightly turbid, not clear enough, the aroma of tea soup was not evident, the tea soup was sour and astringent, and the degree of the leaf base stretch was low.

After sensory evaluation, the results are as shown in Table 1. The scores from high to low are 1 > 2 > 5 > 4 > 3 > 6. It can be seen from the results that freeze-drying can better retain the original aroma and taste of the *Bletilla striata* flowers, while stove-drying will make the sugar aroma in the *Bletilla striata* flowers more prominent and slightly burnt, while air-drying is difficult to retain the aroma and taste of the *Bletilla striata* flowers. In addition, steam fixation will make the sugar in the flower more soluble and make the tea soup sweeter, but it will destroy the fragrance and shape of the flower and affect the appearance [6].

### 2.2. Effect of Different Drying Methods on the Components of Bletilla striata Scented Tea

#### 2.2.1. Analysis of GC-MS

As shown in Table 2, a total of 103 volatile components was detected by GC-MS from the *Bletilla striata* scented tea with different drying methods, including 34 alkanes, 2 olefin, 23 aromatic compounds, 5 esters, 10 alcohols, 13 aldehydes, 8 organic acids, and 8 other types. A total of 48, 55, 55, 51, 57 and 54 volatile components was detected from No. 1 to No. 6. It can be seen that the volatile components of *Bletilla striata* scented tea treated by different drying methods are different to some extent, and *Bletilla striata* scented tea is different from other scented tea in that its hydrocarbon compounds account for a high proportion of the total volatile components, from 19.97 μg/g~39.95 μg/g, and it is found that the mass fraction of hydrocarbon components is generally high after steam fixation, which may be due to the high mass fraction of polysaccharides in *Bletilla striata* flowers, and the high temperature process of steam fixation promotes the decomposition of polysaccharides and aliphatic hydrocarbons to produce more small molecular hydrocarbons [11,12], but it has little contribution to the aroma of *Bletilla striata* scented tea on the whole because the threshold of alkanes is very high.

Aromatic compounds are a special kind of *Bletilla striata* scented tea [13]. Among them, components such as benzenes, dihydrophenanthrene, phenanthrene, and diphenylene are considered as important active components in *Bletilla striata* flowers, and aromatic components are also the main source of the special flavor of *Bletilla striata* flowers. From the results of GC-MS, steam fixation is not conducive to the retention of aromatic compounds, and the stove-drying method has the greatest difference before and after blanching. Ester composition is a representative component in flowers and fruits, which endows flowers and fruits with sweet flavor. Among them, ethyl caproate, the key flavor substance in esters, has the highest mass fraction in freeze-dried *Bletilla striata* scented tea. No ethyl caproate was detected in the No. 3 (air-dried) scented tea, resulting in the loss of this flavor. Alcohols are components with fresh and fruity aromas, among which phenylethanol and *N*-heptanol can provide the special fragrance of *Bletilla striata* scented tea, and phenylethanol is only detected in *Bletilla striata* scented tea without steam fixation. Because phenylethanol is a water-soluble alcohol component, it is speculated that it will be dissolved by a large amount of water and then lost in the process of steam fixation. However, *N*-heptanol was only detected in freeze-dried *Bletilla striata* scented tea. Aldehydes have a high mass fraction in *Bletilla striata* scented tea, among which 2-en-hexaldehyde, nonaldehyde, cinnamaldehyde, *N*-heptaldehyde, etc., provide modifications to the sweet and floral aroma of *Bletilla striata* scented tea. Acids generally have a rancid taste, but their threshold value is generally high, and their impact on the aroma of *Bletilla striata* scented tea is weak. Among the other ingredients, pyrazine was only detected in No. 3 (air-dried), which had an unpleasant and irritating odor.

#### 2.2.2. Analysis of ROAV Results

The ratio of the mass fraction of volatile flavor substances to their threshold value can more accurately reflect the contribution of this substance to aroma [14,15]. The ROAV was used to analyze all flavor substances. The calculation results are shown in Table 3. A total of 22 flavor substances was analyzed. Heptanal, cinnamaldehyde, nonaldehyde, and ethyl caproate were the main key flavor substances, which provided the basic flavor for the *Bletilla striata* flower tea. Among them, heptanal has a similar sweet apricot and nut aroma; cinnamaldehyde has a cinnamon aroma and burning aroma; nonaldehyde provides a citrus aroma, fat aroma, and flower aroma; and ethyl caproate provides a fruit aroma. In addition, 2-en-hexaldehyde was the key flavor substance, except in No. 1 and No. 4, which indicated that stove-drying and air-drying can provide 2-en-hexaldehyde with a grass, fruit, and fat fragrance. However, 1-octen-3-ol was only used as the key flavor substance in No. 5 and No. 6, which indicated that the steam fixation has helped the production of 1-octen-3-ol, which had a green flavor, a vegetable flavor, and a mushroom flavor. The main flavoring substances with modification effects were p-methylphenol, phenylethanol, 2-pentylfuran, and pyrazine. Among them, p-methylphenol had a high mass fraction in the *Bletilla striata* scented tea without steam fixation, providing a tobacco aroma. Phenylethanol was only detected in the *Bletilla striata* scented tea without steam fixation, with the aroma of hyacinth and gardenia, while 2-pentylfuran and pyrazine were mainly detected in samples No. 3 and No. 6, which may produce unpleasant pungent odor.

The greater the ROAV, the greater the contribution to the overall aroma of the sample. The substance with 1 ≤ ROAV is defined as the key aroma substance of the sample, and the substance with 0.1 ≤ ROAV < 1 has an important modifying effect on the overall aroma.

Through the ROAV results, it can be found that the aroma of scented tea is a complex special aroma, in which there are many key flavor substances and flavor substances with modification effects, and different drying methods and steam fixations, or not, have great impacts on the flavor of *Bletilla striata* scented tea.

#### 2.2.3. Mass Fraction Analysis of Total Flavonoids, Total Polysaccharides, and Total Phenols

Flavonoids, sugars, and phenols are generally present in the flower, among which flavonoids, polysaccharides, and total phenols are generally its active components [16,17,18]. According to the results in Figure 2, the total flavonoid content of No. 4 tea is the highest, followed by No. 1 and No. 5 tea, and No. 3 tea is the lowest. The flavonoid components are best preserved in freeze-drying, followed by stove-drying, and the lowest in air-drying. This is because light will decompose the flavonoid components [19], while the air-drying process takes the longest time and the longest exposure time, and the flavonoid content decreases most significantly. Steam fixation can retain more flavonoids. It is speculated that the cell wall after high temperature steam treatment is more fragile after drying during the steaming process, and the flavonoids are more easily dissolved during extraction, thus the mass fraction of flavonoids is higher. When the samples without steam fixation are dried, some enzymes may remain some activity, which will decompose the flavonoids and reduce their mass fraction.

As seen in Figure 2, the polysaccharide content of No. 1 was the highest, followed by No. 2, and the polysaccharide content in the scented tea after steam fixation was generally reduced. However, the polysaccharide content in No. 3 *Bletilla striata* scented tea was the lowest, which may be because the cells and related enzymes of scented tea do not completely lose their activity during the long time of air-drying, thus they will continue to produce respiration and consume the polysaccharide components. Whether freeze-drying or stove-drying, it greatly reduces the enzyme activity and saves a large number of polysaccharide components [20].

As an important effective component in scented tea, total phenols mainly play the role of anti-tumor, anti-oxidation, antibacterial and improving immunity. From the results in Figure 2, it can be seen that the mass fraction of total phenols in *Bletilla striata* scented tea decreased significantly after steam fixation, which may lead to the degradation of total phenols due to the high temperature, as well as the dissolution of some water-soluble phenols after steam fixation. In addition, freeze-drying is the best way to retain total phenols, followed by stove-drying.

### 2.3. Effect of Different Drying Methods on the Efficacy of Bletilla striata Scented Tea

#### 2.3.1. Analysis of Antioxidant Activity

Antioxidant activity is the most common and very important activity of general scented tea, which can delay the aging of human cells and remove free radicals from human body [21,22]. It is an essential activity index of *Bletilla striata* scented tea as a tea drink. From the result of Figure 3, the DPPH clearance rate is consistent with the ABTS clearance ability, which is 1 > 4 > 2 > 5> 3 > 6. This showed that freeze-drying is the best drying method to retain its antioxidant properties, followed by stove-drying and air-drying, with or without steam fixation. This result also suggested that the decomposition and inactivation of the antioxidant active components in the *Bletilla striata* flowers occurred under high temperature during steam fixation. From the perspective of antioxidant activity, direct freeze-drying without steam fixation is the best drying method for *Bletilla striata* scented tea.

#### 2.3.2. Antibacterial Analysis

Scented tea often has the function of clearing heat and detoxifying and is considered to have antibacterial effect in modern medicine [23]. As the most common pathogenic bacteria, *Escherichia coli*, *Pseudomonas aeruginosa*, and *Staphylococcus aureus* were used for the evaluation of the antibacterial properties of *Bletilla striata* flowers treated with different drying methods. As shown in Figure 4, for the inhibition of *Pseudomonas aeruginosa*, No. 1, No. 2, No. 3, and No. 4 had better effects, which may be due to the retained antibacterial substances without steam fixation. However, there was no significant difference in the inhibition of *Staphylococcus aureus* among the six groups, showing that drying methods had little effect on the components of anti-Gram-positive bacteria. For the inhibition of *Escherichia coli*, No. 1 was the best, No. 2, No. 3, No. 5 and No. 6 were the second, and No. 4 was the worst, indicating that freeze-drying can better retain the anti-*Escherichia coli* activity without steam fixation.

## 3. Materials and Methods

### 3.1. Materials and Reagent

Fresh flowers of *Bletilla striata* were harvested from the Traditional Chinese medicine Resource Garden of Crop Research Institute of Guizhou in the middle of May 2022. All flowers were divided into 6 groups (>500 g per group). All chemicals were purchased from Sigma-Aldrich Ltd. unless otherwise stated and used without further purification (analytical level). A DPPH free radical scavenging capacity test kit and an ABTS total antioxidant capacity test kit were purchased from Solarbio Life Science (Beijing, China).

### 3.2. Instrumentation

Volatile substances were determined using 7890A-5975C GC-MS (Agilent Technology Co., Ltd., Santa Clara, CA, USA). Total flavonoids, total polysaccharides, and total phenols were determined using a HP8453 UV spectrophotometer (Aoyi Instruments (Shanghai) Co., Ltd., Shanghai, China). Freeze drying was determined using a CTFD-18S Freeze dryer (Qingdao Yonghe Chuangxin Electronic Technology Co., Ltd., Qingdao, China). Drying was determined using a DHG oven (Shanghai Xinmiao Medical Device Manufacturing Co., Ltd., Shanghai, China).

### 3.3. Treatment of Bletilla striata Flowers

#### 3.3.1. Pre-Treatment

Fresh *Bletilla striata* flowers were rinsed with running water slightly to remove dust and soil and then placed in a cool and ventilated place, and water was drained until there was no evident damp feeling in the hands, numbered 1~6, respectively.

#### 3.3.2. Drying

Put No. 4, No. 5 and No. 6 of the *Bletilla striata* scented tea obtained in 1.3.1 on the steamer that has been aerated, and take it out after 3 min of blanching, while No. 1, No. 2 and No. 3 are not subject to blanching treatment. Freeze No. 1 and No. 4 *Bletilla striata* flowers in a refrigerator at −80 °C overnight, and then freeze them in a freeze dryer for 5 h. Put No. 2 and No. 5 *Bletilla striata* flowers in an oven at 50 °C for 12 h. Lay the No. 3 and No. 6 *Bletilla striata* flowers on the tray, and place them in a ventilated place to dry naturally for 5 days to obtain 6 groups of *Bletilla striata* scented tea, and put them in a refrigerator at −4 °C for cold storage.

#### 3.3.3. Sensory Evaluation

After placing the refrigerated samples at room temperature for 2 h and taking photos, evaluate the taste of the tea soup according to the Methods for Sensory Evaluation of Tea (GB/T 23776-2018). Accurately weigh 3.0 g of flower tea in a 150 mL review cup, add 150 mL of boiling water, cover and brew for 3 min, each 3 times in parallel, after brewing, filter out the tea broth in order of brewing in the evaluation tea bowl. The full score is 100 points, which is divided into five parts: appearance, aroma, taste, soup and leaf bottom evaluation. 100–90 points are excellent, 89–80 points are good, 79–70 points are average, 69–60 points are poor, and less than 60 points are not up to the requirements of tea drinking. The sensory evaluation team is composed of five members who have been trained in tea sensory evaluation.

#### 3.3.4. GC-MS Determination

Place the refrigerated sample at room temperature for 2 h, grind the powder and pass it through a 200-mesh sieve, take a proper amount of sample and transfer it into a 15 mL extraction bottle, and seal it quickly. Age the SPME extraction fiber head to no impurity peak at 240 °C at the GC-MS sample inlet. Place the sample bottle on the solid phase microextraction device, set the temperature at 60 °C, and stir at 500 rpm. Put the sample bottle in the extraction device for preheating for 15 min, insert the SPME extraction head into the headspace part of the sample through the bottle cap, push out the fiber head, the extraction head is about 1.0 cm higher than the upper surface of the sample, and the headspace extraction is 50 min. Pull back the fiber head, pull out the extraction head from the sample bottle, insert the extraction head into the GC-MS sample inlet, push out the fiber head, analyze at 240 °C for 3 min, and then inject the sample for analysis.

Chromatographic conditions: the chromatographic column is DB-WAX (30.0 m × two hundred and fifty μm, 0.25 μm). The initial temperature of the chromatographic column is 40 °C for 2 min, rising to 85 °C at the rate of 3 °C/min for 2 min, rising to 110 °C at the rate of 2 °C/min for 2 min, rising to 160 °C at the rate of 5 °C/min for 1 min, rising to 220 °C at the rate of 5 °C/min for 5 min, gasification chamber temperature 240 °C, transmission line temperature 230 °C, carrier gas He, carrier gas flow rate 1 mL/min, no split flow.

Mass spectrum conditions: EI source, electron energy 70 eV, ion source temperature 230 °C, quadrupole 150 °C, scanning mode Scan, scanning mass range 35 u~500 u.

Qualitative analysis: MS database NIST11 and retention time are used for qualitative analysis of the detected components. The column loss peak should be deducted from the database screening results.

Quantitative analysis: *N*-decanol with a concentration of 100 μg/mL was used as the internal standard. The mass concentration of each component can be obtained by calculating the ratio of the peak area of each component to the peak area of the internal standard, then multiplying it with the mass concentration of the internal standard. The specific formula is as follows:(1)$$\mathrm{Mass}\;\mathrm{concentration}\;\mathrm{of}\;\mathrm{volatile}\;\mathrm{components}\left(\frac{\mu g}{g}\right)\;=\frac{\mathrm{Compositional}\;\mathrm{peak}\;\mathrm{area}\times \mathrm{Mass}\;\mathrm{concentration}\;\mathrm{of}\;\mathrm{internal}\;\mathrm{standard}\;\mathrm{material}\left(\mu g/g\right)}{\mathrm{Internal}\;\mathrm{standard}\;\mathrm{peak}\;\mathrm{area}}$$

The NIST database is used to search, detect impurity peaks such as polyoxysilane (column loss component), and use the internal standard semi-quantitative method to carry out the quantitative results.

#### 3.3.5. ROAV Analysis

ROAV was used to analyze the contribution of each volatile flavor substance to the overall aroma, and the odor activity value (OAV) was used to determine the substance that contributed the most to the aroma [24,25], which can be calculated as follows:OAV = C/T(2)
where C and T are the mass fraction and the sensory threshold of a substance (μg/kg).

According to the results of OAV, the component that contributes the most to the aroma is defined as ROAVmax = 100, and other flavor substances are calculated by Equation (3):(3)$$ROAV_{i}\approx \;100\times \frac{C_{i}}{C_{max}}\times \frac{T_{max}}{T_{i}}$$
where $ROAV_{i}$ represents the relative odor activity value of each substance; $C_{i}$ and $C_{max}$ indicates the mass fraction of each substance and the mass fraction of $ROAV_{max}$ substance (μg/g); $T_{i}$ and $T_{max}$ are the sensory threshold of each substance and the sensory threshold of $ROAV_{max}$ substance (μg/kg).

#### 3.3.6. Determination of Total Flavonoids, Total Polysaccharides and Total Phenols

Put the refrigerated sample at room temperature for 2 h, beat the powder and pass it through a 200-mesh sieve, weigh 1.0 g accurately, add 20 mL of 80% ethanol, extract for 30 min by ultrasound, centrifuge and filter the residue to obtain the test solution, and determine the mass fraction of total flavanone according to the method in the Determination of Total Flavonoids in Exported Foods (SN/T 4592-2016). The mass fraction of the total polysaccharides shall be determined according to the method of the Phenol-Sulfuric Acid Method for the Determination of Crude Polysaccharides in Plant Derived Foods for Export (SN/T 4260-2015), and the mass fraction of total phenols shall be determined according to the method of Zhang S. and Han X. [26].

#### 3.3.7. Determination of Oxidation Resistance

Leave the refrigerated sample at room temperature for 2 h, beat the powder and pass it through a 200-mesh sieve, weigh 0.2 g accurately, add 10 mL of deionized water, ultrasonic extraction for 30× *g* min, centrifuge, filter the residue to obtain 20 mg/mL of test solution, and then use the dilution method to obtain 10, 5, 2, 1, and 0.5 mg/mL of test solution, and determine according to the methods described in the instructions of the total antioxidant capacity kit (DPPH method) and the total antioxidant capacity kit (ABTS method).

#### 3.3.8. Determination of Antibacterial Ability

Using *Escherichia coli*, *Staphylococcus aureus*, and *Pseudomonas aeruginosa* as the test bacteria, the antibacterial effects of six samples were determined by the filter paper agar diffusion method and the co-culture method. *Escherichia coli*, *Staphylococcus aureus*, and *Pseudomonas aeruginosa* grow on the nutrient agar plate and can be stored for a period of time at 4 °C. When necessary, take the agar plate to room temperature, and rewarm it in a sterile environment for 40 min, then use the inoculation ring beside the flame to move a single bacterial colony into the LB broth, and shake it for 12 h at 37 °C to obtain the bacterial suspension, and compare it with the 0.5 # Maxwell tube (1.5 × 10^8^ cfu/mL) dilute the above LB broth with normal saline to a similar turbidity, and then obtain a suspension of 108 bacteria. Weigh 10 mg of each sample and immerse it in 1 ml of PBS for 24 h.

Immerse the sterile filter paper (diameter 6 mm) in the extract of two samples for 30 min, and absorb the above concentration of about 1 × 10^8^ cfu/mL of three bacterial suspensions, each 200 μL. Apply evenly on the nutrient agar plate, place the soaked sterile filter paper evenly on the agar plate, take PBS as the negative control, then incubate the plate upside down at 37 °C for 24 h, and record the diameter of the bacteriostatic ring.

#### 3.3.9. Statistical Analysis

The experiments were repeated three times, and the data were analyzed with the SPSS V22.0 software and plotted using origin 2018.

## 4. Conclusions

This study studied *Bletilla striata* scented tea using sensory evaluation under different drying methods. Different drying methods indeed affected the shape, color, aroma, composition and efficacy of *Bletilla striata* scented tea. It was found that *Bletilla striata* scented tea had the highest score under the freeze-drying method without steam fixation. In addition, it was found that freeze-dried flower tea could better retain the original aroma and flavor of *Bletilla striata* flowers by making flowers slightly scorched. Through the analysis of the components in the scented tea, it was found that the aroma of the scented tea was a complex special aroma, in which there were many key flavor substances. The steam fixation and air-drying were not conducive to the retention of aromatic compounds, and, particularly, air-drying would lead to the loss of key flavor substances such as ethyl caproate. By contrast, the freeze-drying method benefits the retention of flavonoids, polysaccharides, and total phenols. From the perspective of efficacy, the freeze-drying method without steam fixation showed good effects in both antioxidant and antibacterial properties. In general, freeze-dried tea can better retain the flavor and effective components of *Bletilla striata* scented tea, and can provide better antioxidant and antibacterial effects. Stove-drying slightly changes the aroma of *Bletilla striata* scented tea to produce other flavors, which is more economical and convenient in actual production and application. Air-drying will reduce the quality of *Bletilla striata* scented tea. Steam fixation destroys the form and flavor of *Bletilla striata* scented tea, changes the effective components of *Bletilla striata* scented tea, and reduces the efficacy of *Bletilla striata* scented tea. In summary, direct freeze-drying without steam fixation is the best process for drying *Bletilla striata* scented tea, stove drying without steam fixation is more economical and convenient in actual production and application, and steam fixation and air-drying are not suitable as drying processes for *Bletilla striata* scented tea.

## Figures and Tables

**Figure 1 molecules-28-02438-f001:**
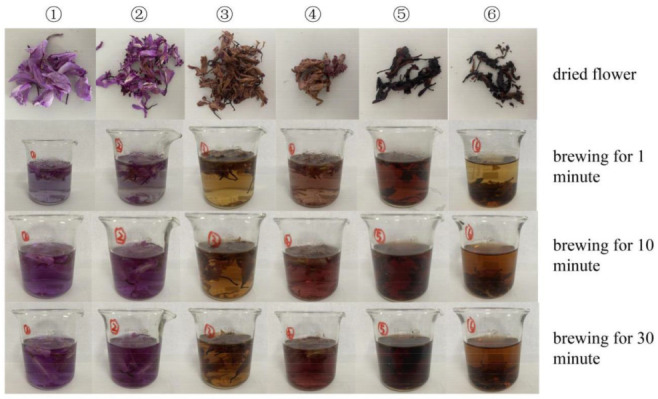
*Bletilla Striata* Scented Tea and Brewing Effect.

**Figure 2 molecules-28-02438-f002:**
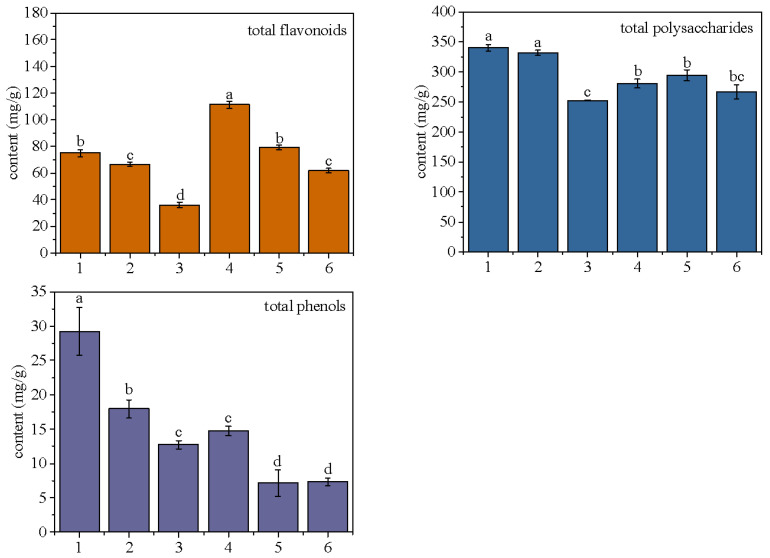
Mass fraction of *Bletilla striata* flowers mosaic flavonoids, polysaccharides, and total phenols. Note: Different alphabets indicate statistical differences between groups, and the same alphabets indicate no statistical differences between groups.

**Figure 3 molecules-28-02438-f003:**
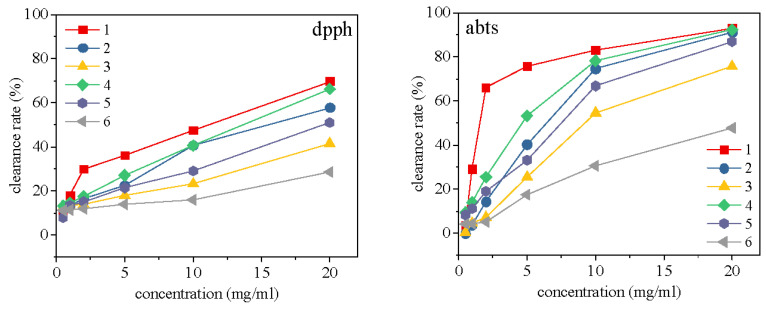
Antioxidant of flowers.

**Figure 4 molecules-28-02438-f004:**
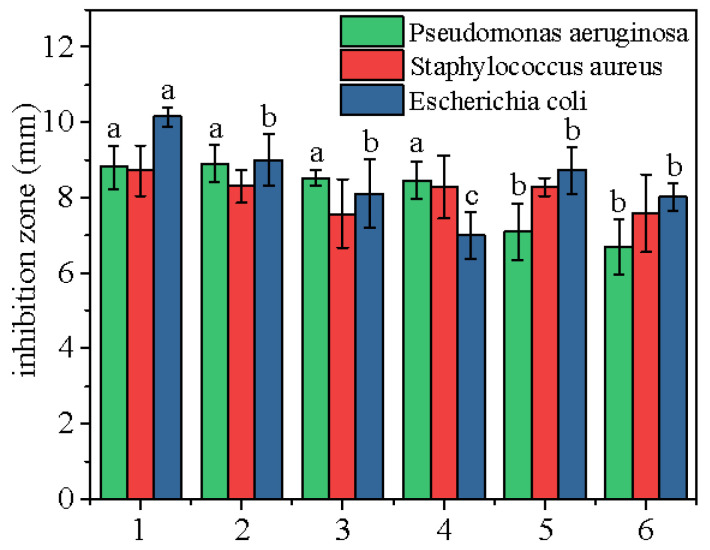
Antibacterial activity of *Bletilla striata* flowers. Note: Different alphabets indicate statistical differences between groups, and the same alphabets indicate no statistical differences between groups.

**Table 1 molecules-28-02438-t001:** Sensory evaluation results of *Bletilla striata* flowers.

Part		Number					
		1	2	3	4	5	6
Appearance	result	The flower shape is complete, full, and purple in color	The flower shape is more complete, fuller, color purplish red with some brown	The flowers slightly crushed, fuller brown color with purplish red	The flower shape is slightly broken, more crumpled, the color is slightly purplish red and slightly white	The flower shape is not clear, wrinkled, dark brown color	The flower shape is not clear, more broken, crumpled, dark brown color
	score	90	88	80	80	72	72
Aroma	result	The unique fragrance and rich fragrance of the *Bletilla striata* flowers	Unique fragrance of *Bletilla striata* flowers, slightly burnt	Inconspicuous aroma	Unique fragrance of *Bletilla striata* flowers, Slightly lighter	Caramel aroma, slightly sour	Inconspicuous aroma
	score	90	88	75	78	90	70
Soup	result	The soup is purplish red, clear, and still bright after a long time	The Soup color is purple, clear, long bubble slightly turbid	The color of the soup is brown, clear, and bright	Soup brown, red, clear, long bubble, slightly turbid	The color of the soup is dark brown–red, not clear enough and cloudy for a long time	Brown soup, clear, long bubble slightly turbid
	score	90	86	88	85	80	84
Taste	result	Mellow, sweet, sweet, good coordination	Mellow, sweet, back to sweet, slightly light, good coordination	The taste is mellow, slightly sweet and moderately harmonious	Mellow taste, slightly sweet slightly astringent, general coordination	Mellow taste, mild caramel sweet, moderate coordination	Coarse light, slightly watery, sour taste, no sense of hierarchy
	score	90	88	80	82	85	70
Leaf bottom evaluation	result	Natural stretch, lavender, even	Natural stretch, lavender red, uniform	Slightly stretched, light brown, uniform	Slightly stretched, light reddish brown, more uniform	Slight stretch, black and red, uneven	Slight stretch, black and red, uneven
	score	90	90	84	83	75	75
Score ^a^		90.00	88.2	78.65	80.75	83.15	70.90

^a^: According to the evaluation method of flower tea in GB/T 23776-2009 “Sensory Evaluation Method for Tea”, the score of appearance accounted for 20%, aroma accounted for 35%, soup color accounted for 5%, taste accounted for 30%, and leaf base accounted for 10%. The total score was calculated by the weighted method.

**Table 2 molecules-28-02438-t002:** GC-MS results of *Bletilla striata* flowers.

No.	Compound	Retention Time (min)	Relative Content (%)					
			1	2	3	4	5	6
1	Alkane							
2	10-Methyl decadecane	9.84	-	0.27	-	-	-	-
3	2,5-dimethylnonane	0.99	-	1.02	1.01	1.18	1.45	1.26
4	2,6,10,14-tetramethyl hexadecane	12.95	0.59	-	-	4.22	5.41	-
5	2,6,10-trimethylundecanone	11.41	-	-	-	0.04	0.85	-
6	2,6-dimethyldecane	3.69	-	-	2.83	1.09	-	-
7	2,6-dimethylundecanone	8.12	-	1.75	1.91	1.94	1.25	1.28
8	2-Methylundecanone	4.57	-	-	-	-	0.36	0.36
9	3,6-dimethyldecane	1.76	2.15	5.88	-	6.03	2.55	2.69
10	3,6-dimethylnonane	1.76	2.15	5.88	-	6.03	2.55	2.69
11	3,8-dimethyldecane	18.26	-	0.03	1.19	-	-	-
12	4,6-dimethyldodecane	9.98	0.64	0.24	0.21	0.60	0.44	0.69
13	4,6-dimethyldodecane	8.59	1.05	-	1.47	-	0.92	1.04
14	4,8-dimethyldecane	8.59	1.05	-	1.47	-	0.92	1.04
15	4,8-dimethylundecanone	2.26	1.66	2.51	2.81	3.24	3.68	3.15
16	4-Methyldecane	9.13	-	1.15	-	1.39	-	-
17	4-Methyldodecane	6.07	-	-	0.20	0.25	-	-
18	4-Methylundecanone	6.07	-	-	-	-	0.19	0.2
19	5-Butyl nonane	2.14	0.61	0.75	0.64	0.74	0.78	0.77
20	5-Methyldecane	10.36	-	-	2.27	2.36	-	-
21	5-Ethyl-2-methyl-octane	10.36	-	-	-	-	-	2.07
22	8-methylheptadecane	4.85	-	-	-	-	0.64	-
23	Methyl tris (trimethylsilyl) silane	4.56	-	-	-	0.44	-	0.03
24	*N*-eicosane	3.86	2.75	4.39	5.42	5.41	6.68	6.51
25	*N*-decane	19.43	-	-	-	0.20	0.04	0.2
26	*N*-octadecane	1.89	2.74	0.27	3.39	2.14	2.25	1.99
27	*N*-dodecane	9.42	0.73	0.53	0.71	0.10	1.11	0.82
28	*N*-decadecane	18.28	0.36	0.03	-	-	-	-
29	*N*-hexadecane	9.83	0.15	-	-	-	0.15	-
30	*N*-heptadecane	6.25	-	1.48	0.06	0.40	-	-
31	*N*-tridecane	9.32	-	0.91	0.05	1.48	1.03	0.38
32	Tetradecane	9.05	3.28	1.05	0.31	0.40	0.41	0.28
33	*N*-pentadecane	11.71	-	0.05	-	0.27	0.47	0.49
34	*N*-undecanone	17.64	0.06	-	-	-	-	-
35	Olefin							
36	1,2,3,4-tetramethyl-5-methylene	56.17	1.41	0.18	0.31	-	-	-
37	Transsqualene	17.22	0.76	1.39	-	0.89	-	-
38	Aromatic group							
39	1,2,3,4-tetramethylbenzene	16.78	0.77	-	-	-	-	-
40	1,2,3,5-tetramethylbenzene	20	-	0.09	0.56	-	0.19	0.21
41	1,2,4,5-tetramethylbenzene	14.45	-	0.22	0.60	0.22	0.85	0.07
42	1,3-dimethyl-2-ethylbenzene	20.33	0.20	-	-	-	-	0.11
43	1,3-dimethyl-4-ethylbenzene	38.8	0.09	0.03	-	-	0.05	-
44	1,3-dimethyl-5-ethylbenzene	20.33	-	-	-	-	0.15	-
45	1,4-dihydro-1,4-methylbridgenaphthalene	22.24	0.05	-	0.09	-	0.05	-
46	1-ethyl-2-isopropylbenzene	21.98	-	-	-	0.22	-	0.12
47	2,3-dichlorotoluene	37.6	-	0.06	0.06	0.04	-	0.05
48	2,5-dichloromethylbenzene	20.33	-	-	0.19	-	-	-
49	2-methylnaphthalene	21.97	-	0.14	-	0.09	-	0.22
50	2-tert-butyl toluene	20.33	0.20	-	-	-	-	0.11
51	3,4-dichlorotoluene	21.97	-	-	-	-	0.21	0.05
52	3,5-dimethyl-1-isopropylbenzene	13.84	0.05	-	0.05	-	0.05	-
53	3,5-dichlorotoluene	27.61	-	-	0.02	-	-	-
54	3-ethylo-xylene	19.64	-	-	-	-	-	0.45
55	4-(2-methyl-2-propenyl) phenol	13.86	-	0.06	0.20	-	-	0.06
56	4-isopropyltoluene	42.98	0.10	0.08	-	0.02	0.03	-
57	5-Ethyl-3,5-dimethylbenzene	44.92	1.50	8.22	0.17	0.19	0.06	-
58	phenol	19.64	-	0.67	-	-	0.07	-
59	*P*-methylphenol	32.25	0.58	0.57	-	0.37	0.43	0.4
60	*O*-isopropyl methylbenzene	27.61	-	0.19	-	0.20	-	0.08
61	naphthalene	56.67	-	-	-	-	0.13	-
62	Pentamethylbenzene	26.62	-	-	1.43	-	-	-
63	Esters							
64	Dibutyl 1,2-phthalate	60.39	1.18	-	-	-	-	-
65	Acetyl 4-hydroxybutyrate	49.58	0.10	0.06	-	0.18	-	0.14
66	Ethyl caproate	19.01	1.14	0.51	-	1.00	0.57	0.99
67	Di (2-ethylhexyl) phthalate	23.62	-	-	0.23	-	-	1.94
68	Dimethyl phthalate	21.43	-	1.16	-	-	-	-
69	Alcohols							
70	1-octene-3-ol	29.76	-	-	-	-	0.24	0.08
71	2,3-Butanediol	43.96	-	-	-	-	-	0.11
72	α-Cyanobenzyl alcohol	40.38	1.90	0.24	0.25	1.15	0.67	1.37
73	Benzyl alcohol	49.4	0.16	-	-	0.04	0.04	0.02
74	Furfuryl alcohol	19.25	-	-	-	-	-	0.29
75	Hydrocinnitol	14.64	-	-	0.16	-	-	2.33
76	Phenylethanol	36.25	2.61	1.84	1.19	-	-	-
77	Cinnamyl alcohol	43.23	-	0.02	0.04	-	0.41	-
78	*N*-heptanol	5.62	-	-	-	0.48	-	-
79	*N*-hexanol	36.26	-	-	-	1.24	-	0.68
80	Aldehyde							
81	2,4-Dimethylbenzaldehyde	21.72	1.69	0.40	1.17	1.54	1.37	1.74
82	2-pyrrolaldehyde	43.47	0.05	-	-	0.06	-	-
83	2-en-hexaldehyde	18.82	-	0.55	0.98	-	7.22	1.31
84	3,4-dimethylbenzaldehyde	22.59	0.13	-	-	-	-	0.48
85	3,5-dimethylbenzaldehyde	16.04	-	0.11	0.93	0.40	0.15	-
86	Benzaldehyde	43.45	-	-	0.05	-	-	-
87	Transcinnamaldehyde	54.37	-	-	0.05	-	-	-
88	Furan formaldehyde	6.86	2.20	2.50	7.68	3.07	3.53	2.25
89	Trans-2-nonenal	52.13	0.22	0.09	0.10	0.12	0.12	0.07
90	Nonanal	56.81	0.05	0.94	0.08	0.17	0.18	-
91	Cinnamaldehyde	18.51	1.26	3.95	0.95	2.89	4.77	2.98
92	Vanillin	58.73	1.81	-	0.02	-	-	0.07
93	Heptaldehyde	27.95	6.28	0.59	0.32	0.49	0.37	0.41
94	Organic acid							
95	Benzoic acid	38.63	2.63	0.85	2.87	1.37	0.19	1.04
96	Myristic acid	61.12	0.49	0.89	0.24	1.01	0.53	0.15
97	acetic acid	20.13	0.19	-	0.16	-	-	-
98	stearic acid	8.89	-	0.09	0.59	-	-	0.25
99	*N*-butyric acid	20.85	-	0.16	-	-	0.91	-
100	Heptanoic acid	41.98	-	0.06	-	-	0.12	-
101	Hexanoic acid	7.84	-	-	0.19	-	-	-
102	palmitic acid	12.95	0.59	-	-	4.22	5.41	-
103	Other							
104	2-methylindene	25.03	2.96	2.08	1.00	1.38	1.03	-
105	2-pentylfuran	8.89	-	0.09	0.59	-	-	0.25
106	2-Acetylfuran	20.85	-	0.16	-	-	0.91	-
107	2-Acetylpyrrole	43.23	-	0.02	0.04	-	0.41	-
108	Pyrazine	7.84	-	-	0.19	-	-	-
109	*N*,*N*-dimethylformamide	12.95	0.59	-	-	4.22	5.41	-
110	Chamomile	32.24	-	-	0.50	-	-	-
111	Isophorone	25.03	2.96	2.08	1.00	1.38	1.03	-

**Table 3 molecules-28-02438-t003:** ROAV analysis of *Bletilla striata* flowers.

Compounds	Threshold	ROAV ^a^					
		1	2	3	4	5	6
phenol	5900	- ^b^	0.01	-	-	0.00 ^c^	-
*P*-methylphenol	90	0.31	0.30	-	0.20	0.23	0.21
Ethyl caproate	1	54.46	24.36	-	47.77	27.23	47.29
1-octene-3-ol	1	-	-	-	-	11.46	3.82
Furfuryl alcohol	5000	-	-	-	-	-	0.00
Phenylethanol	800	0.16	0.11	0.07	-	-	-
Cinnamyl alcohol	40	-	0.02	0.05	-	0.49	-
*N*-heptanol	3	-	-	-	7.64	-	-
*N*-hexanol	2500	-	-	-	0.02	-	0.01
2-en-hexaldehyde	17	-	1.55	2.75	-	20.29	3.68
Benzaldehyde	1500	-	-	0.00	-	-	-
Nonanal	1	2.39	44.90	3.82	8.12	8.60	-
Cinnamaldehyde	14	4.30	13.48	3.24	9.86	16.28	10.17
Vanillin	32	2.70	-	0.03	-	-	0.10
Heptaldehyde	3	100.00	9.39	5.10	7.80	5.89	6.53
Myristic acid	10,000	0.00	0.00	0.00	0.00	0.00	0.00
acetic acid	22,000	0.00	-	0.00	-	-	-
*N*-butyric acid	6500	-	0.00	-	-	0.01	-
2-pentylfuran	6	-	0.72	4.70	-	-	1.99
2-Acetylfuran	10,000	-	0.00	-	-	0.00	-
2-Acetylpyrrole	170,000	-	0.00	0.00	-	0.00	-
Pyrazine	60	-	-	0.15	-	-	-

^a^: For odor threshold, please refer to references (Wine flavor chemistry, Food flavoring technique); ^b^: “-” means not detected; ^c^: “0.00” means that the ROAV value of the sample is lower than 0.01.

## Data Availability

The datasets used or analyzed during the current study are available from the corresponding author upon reasonable request.

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
