# Peer review of "Effects of Different Drying Methods on the Quality of *Bletilla striata* Scented Tea"

_molecules, 2023, doi:10.3390/molecules28062438_

Round 1
Reviewer 1 Report
1. In the Abstract, the detailed results of research should be included, such as the main aroma compounds, the content, and critical aromas’ ROAV. Now, there are too many descriptive words for Abstract.
2. All of Latin species name should be italic. How many authentic standards were used in the identification of Aroma compounds?
3. 2.3.3 actually, the standard is used for Tea (Camellia species), so if it could be applied to other herbal teas? And the most important factor is the panelist, how many people and the training procedure should also be introduced.
4. Line 131, this line is not complete.
5. In Table 2, how about the SD?
Author Response
< Molecules>
Title: Effects of different drying methods on the quality of Bletilla striata scented tea
Ref. No.: molecules-2236032
Dear editors and reviewers,
On behalf of my co-authors, we thank you very much for giving us an opportunity to revise our manuscript; we appreciate the editor and reviewers very much for their positive and constructive comments and suggestions on our manuscript.
We have revised our manuscript according to the comments. The revised parts in the manuscript (including words, sentences and paragraphs) have been highlighted by a red color. Attached please find the revised manuscript, which we would like to submit for your kind consideration.
Looking forward to hearing from you.
Thank you and best regards.
Answer sheet for the reviewers’ comments
Reviewer #1: In the Abstract, the detailed results of research should be included, such as the main aroma compounds, the content, and critical aromas’ ROAV. Now, there are too many descriptive words for Abstract.
Answer to this comment:
Following the reviewers comment, modified the following to the revised manuscript.
lines 12-27: The findings show that the freeze-drying method can retain the original aroma and flavor of Bletilla striata has the highest sensory evaluation score, with the key flavor substances ethyl caproate and n-heptanal containing 1.14% and 6.28%, respectively, and their ROAV values reaching 54.46 and 100.00. And, the freeze-drying method can well retain flavonoids, polysaccharides and phenolic components, while providing better antioxidant and antibacterial properties. The stove drying method would make Bletilla striata slightly burnt and less flavorful and efficacious than freeze-drying; the air-drying method is difficult to retain the special odor and fragrance of Bletilla striata flowers and has the lowest sensory evaluation score, with the presence of volatile components with irritating and unpleasant odors such as pyrazine and 2-pentylfuran, while not showing better efficacy. In addition, the steam fixation would destroy the morphology and flavor of Bletilla striata, lose polysaccharide and phenolic components, and reduce the efficacy of Bletilla striata scented tea, but could retain the flavonoid components well. In summary, direct freeze-drying without steam fixation is the best process for drying Bletilla striata scented tea, stove drying without steam fixation is more economical and convenient in actual production and application, steam fixation and air-drying are not suitable as drying process for Bletilla striata scented tea.
Reviewer #2:All of Latin species name should be italic. How many authentic standards were used in the identification of Aroma compounds?
Answer to this comment:
The Latin names that appear have been italicized as required. Since this paper uses a semi-quantitative method, the detected components were identified qualitatively using the MS database NIST11, retention time, and those with more than 90% match were taken for analysis.
Reviewer #3:2.3.3 actually, the standard is used for Tea (Camellia species), so if it could be applied to other herbal teas? And the most important factor is the panelist, how many people and the training procedure should also be introduced.
Answer to this comment:
The scope of application of the standard is all kinds of tea, which includes this type of herbal tea.
Following the reviewers comment, add the following to the revised manuscript.
Line 98:accurately weigh 3.0 g of flower tea in a 150 mL review cup, add 150 mL of boiling water, cover and brew for 3 min, each bubble 3 parallel, after brewing, filter out the tea broth in order of brewing in the evaluation tea bowl .
Reviewer #4: Line 131, this line is not complete.
Answer to this comment:
Line 131:Already modified ”with mass concentration of 100 μg/mL of n-decanol as the internal standard.”to ”N-decanol with concentration of 100 μg/mL was used as internal standard.”
Reviewer #5: In Table 2, how about the SD?
Answer to this comment:
Due to the accuracy of the GC-MS results, the SD in the data obtained are very minimal, so that the results are without marking SD and show only the average of three times.

Reviewer 2 Report
The manuscript reported the effects of different drying methods on the quality of Bletilla striata scented tea. The experiments were systematically designed and the paper is well organized. The results can provide important reference for industrial production. In this respect, the manuscript is suggested to be accepted in Molecules after minor revision. The followings are some comments and suggestions for authors to consider and improve the manuscript.
[1] Line 3: replace the words "baiji" with "Bletilla striata"
[2] Lines 89-96: How to make sure the Bletilla striata flowers are completely dry?
[3] Line 152: Are total flavonoids, total polysaccharides and total phenols all extracted with deionized water? How can they be completely extracted?
[4] Lines 170-172: Why are these three types of bacteria used?
[5] Line 213: The title of Fig. 1 should be revised.
[6] Line 229: The amounts of volatile components do not add up to 111
[7] Line 243: The first row of Table 2 should be underlined
[8] Line 266: In terms of the relationship between the value of the ROAV and the overall aroma contribution of the sample, how to distinguish the key aroma substances and the important modification substances?
[9] Line 381: Authors are suggested to summarize the effects of different drying methods on flower tea from different angles.
[10] There are some errors for the reference style, please revise it according to the guide for authors of the journal.
[11] Line 455, Camellia sinensis, line 422, Hosta plantaginea, line 412, 415, Bletilla striata should be italic.
Author Response
< Molecules>
Title: Effects of different drying methods on the quality of Bletilla striata scented tea
Ref. No.: molecules-2236032
Dear editors and reviewers,
On behalf of my co-authors, we thank you very much for giving us an opportunity to revise our manuscript; we appreciate the editor and reviewers very much for their positive and constructive comments and suggestions on our manuscript.
We have revised our manuscript according to the comments. The revised parts in the manuscript (including words, sentences and paragraphs) have been highlighted by a red color. Attached please find the revised manuscript, which we would like to submit for your kind consideration.
Looking forward to hearing from you.
Thank you and best regards.
Answer sheet for the reviewers’ comments
Reviewer #1:Line 3: replace the words "baiji" with "Bletilla striata"
Answer to this comment:
Line 39:Already modified "baiji" to "Bletilla striata" .
Reviewer #2:Lines 89-96: How to make sure the Bletilla striata flowers are completely dry?
Answer to this comment:
The flowers are very easy to dry, and the above drying time has been verified in previous tests to be sufficient to completely dry the Bletilla striata flower.
Reviewer #3:Line 152: Are total flavonoids, total polysaccharides and total phenols all extracted with deionized water? How can they be completely extracted?
Answer to this comment:
Line:Writing errors, modified "deionized water" to "80% ethanol"
Reviewer #4:Lines 170-172: Why are these three types of bacteria used?
Answer to this comment:
Because these are the three most common bacteria that infect people, they are often used as an evaluation indicator of antibacterial activity.
Reviewer #5:Line 213: The title of Fig. 1 should be revised.
Answer to this comment:
Line 213: Already modified "Bletilla striata flowers and brewing effect." to "Bletilla striata scented tea and brewing effect."
Reviewer #6:Line 229: The amounts of volatile components do not add up to 111.
Answer to this comment:
Line 229: Writing errors,Already modified“a total of 111 volatile components were detected by GC-MS from the Bletilla striata scented tea with different drying methods, including 32 alkanes,2 alkanes, 24 aromatic compounds, 5 esters, 10 alcohols, 13 aldehydes, 8 organic acids and 8 other types. ”to“As shown in Table 2, a total of 103 volatile components were detected by GC-MS from the Bletilla striata scented tea with different drying methods, including 34 alkanes, 2 olefin 23 aromatic compounds, 5 esters, 10 alcohols, 13 aldehydes, 8 organic acids and 8 other types.”
Reviewer #7:Line 243: The first row of Table 2 should be underlined.
Answer to this comment:
Line 243: The format of Table 2 has been modified as required.
Reviewer #8: Line 266: In terms of the relationship between the value of the ROAV and the overall aroma contribution of the sample, how to distinguish the key aroma substances and the important modification substances?
Answer to this comment:
Line 290:Already added "The greater the ROAV, the greater the contribution to the overall aroma of the sample. The substance with 1 ≤ ROAV is defined as the key aroma substance of the sample, and the substance with 0.1 ≤ ROAV<1 has an important modifying effect on the overall aroma."
Reviewer #9: Line 381: Authors are suggested to summarize the effects of different drying methods on flower tea from different angles.
Answer to this comment:
Line 408:Already added “In summary, direct freeze-drying without steam fixation is the best process for drying Bletilla striata scented tea, stove drying without steam fixation is more economical and convenient in actual production and application, steam fixation and air-drying are not suitable as drying process for Bletilla striata scented tea.”
Reviewer #10: There are some errors for the reference style, please revise it according to the guide for authors of the journal.
Answer to this comment:
All references have been checked for format and revised according to the submission guidelines. Some references are published online, so page numbers are not available or only codes are available.
Reviewer #11:Line 455, Camellia sinensis, line 422, Hosta plantaginea, line 412, 415, Bletilla striata should be italic.
Answer to this comment:
The Latin names appearing above have been italicized as required.
Round 2
Reviewer 1 Report
After revision, i have no more comments.